# A Lighting Consistency Technique for Outdoor Augmented Reality Systems Based on Multi-Source Geo-Information

Kunpeng Zhu [1,2], Shuo Liu [1,*], Weichao Sun [1], Yixin Yuan [1] and Yuang Wu [1,2]

1    National Engineering Research Center for Geomatics (NCG), Aerospace Information Research Institute,
     Chinese Academy of Sciences, Beijing 100101, China; zhukunpeng211@mails.ucas.ac.cn (K.Z.);
     sunwc@aircas.ac.cn (W.S.); yuanyx@aircas.ac.cn (Y.Y.); wuyuang20@mails.ucas.ac.cn (Y.W.)
2    University of Chinese Academy of Sciences, Beijing 100049, China
*    Correspondence: liushuo@aircas.ac.cn

**Abstract:** Achieving seamless integration between virtual objects and real scenes has always been an important issue in augmented reality (AR) research. To achieve this, it is necessary to provide virtual objects with real-time and accurate lighting conditions from a real scene. Therefore, the purpose of this study is to realize lighting consistency rendering for real-time AR systems in outdoor environments, aiming to enhance the user's sense of immersion. In this paper, we propose a lighting consistency technique for real-time AR systems in outdoor environments based on multi-source geographical information (MGI). Specifically, we introduce MGI into the study of lighting consistency and construct a comprehensive database to store and manage the acquired MGI data. Based on this, we proposed a sky radiance model driven using the MGI. Finally, we utilized the sky radiance model along with light sensor data to render the virtual objects in outdoor scenes. The experimental results show that the shadow angular error is reduced to 5.2°, and the system frame rate is increased to 94.26. This means that our method achieves a high level of realism in the fusion of virtual objects and real scenes while ensuring a high frame rate in the system. With this technology, users can conveniently and extensively realize the lighting consistency rendering of real-time AR systems in outdoor scenes using mobile devices.

**Keywords:** GIS; multi-source geo-information; augmented reality; light consistency; real-time

## 1. Introduction

Augmented reality (AR) refers to a technology that enhances users' perception and understanding of the real world and virtual information by integrating virtual information with real scenes [1]. Achieving realistic integration of virtual objects into real scenes is a popular research field in AR. To achieve this goal, it is necessary to ensure that AR systems have good geometric consistency, temporal consistency (real-time performance), and lighting consistency [2]. Geometric consistency refers to the accurate spatial properties of virtual objects in real scenes, such as their precise position and occlusion relationship. Temporal consistency refers to the synchronization between virtual objects and real scenes, enabling users to interact in real-time. Lighting consistency refers to the similarity in lighting between the rendered scene of virtual objects and the real scene so that virtual objects have realistic lighting effects such as brightness, reflection, and shadows similar to those in the real world. Currently, with the development of marker-based 3D registration algorithms [3,4] and 3D registration algorithms combined with SLAM (simultaneous localization and mapping) [5–7], the problem of geometric consistency has been well solved. The continuous improvement in computer hardware and mobile device processing power also provides possibilities for solving the temporal consistency problem of AR systems. However, estimating the lighting parameters of real scenes remains challenging due to the complexity of the lighting environment [8]. Therefore, further research is needed to address the lighting consistency problem of AR systems.

Augmented reality (AR) is widely used in outdoor environments, such as virtual Olympics, traffic simulation, and AR navigation [9]. In these applications, identifying how to estimate lighting conditions in the outdoor environment to achieve good lighting consistency is a crucial issue for improving user immersion. Currently, mainstream research methods for this problem can be divided into three categories: methods based on marker information [10–12], methods based on auxiliary equipment [13–16], and methods based on image analysis [17–20]. Methods based on marker information and methods based on auxiliary equipment require special devices that collect prior information, such as light probes [10] or fisheye cameras [13], making them difficult to implement on mobile devices. Methods based on image analysis are less restricted by application scenarios but may not accurately estimate lighting conditions in low-information outdoor environments [17–19]. Moreover, these methods have high computational costs and cannot meet the real-time requirements of AR interaction scenes [17–20]. Therefore, in this work, we aim to develop a lighting consistency technique for real-time AR systems in outdoor environments without requiring additional devices or visual information. Furthermore, the technology should be widely applicable to general mobile devices.

Multi-source geographic information data can describe outdoor scenes from different perspectives [21]. Utilizing different geographic environmental information enables a more accurate estimation of lighting parameters of outdoor scenes. Moreover, multi-source geographic information data can be acquired in real-time, providing a basis for meeting the real-time requirements of AR systems. Therefore, we introduce multi-source geographic information into the study of lighting consistency and propose a lighting consistency technique for outdoor real-time AR systems based on multi-source geographic information.

In this paper, the multi-source geographic information in different outdoor scenes is used to obtain the lighting parameters of the scenes, which can realize the consistent lighting rendering of the real-time AR system in outdoor scenes without the need to prepare markers and acquire scene images. Specifically, this paper accurately estimates the position of the sun using geographical location information such as latitude, longitude, altitude, and time obtained from mobile sensors. At the same time, temperature, air pressure, relative humidity, and other geographical environment data are used to calculate the atmospheric turbidity in the user's location. The above results are combined with surface reflectance data from remote sensing to generate a high dynamic range (HDR) sky radiance model for the scene. Then, image-based lighting (IBL) technology is used to generate the ambient light in the scene. Finally, we adjust the solar illumination parameter using the light sensor in a mobile phone to achieve realistic virtual–real integration rendering. This method achieves consistency in lighting in AR systems while meeting real-time requirements. And it can be conveniently implemented on mobile terminals such as mobile phones and tablets.

The contributions of this work can be summarized in three aspects:

Firstly, we introduce multi-source geographic information data and establish a sky radiation model driven using multiple geographic information, which extends previous research methods for addressing the problem of lighting consistency in augmented reality.

Secondly, we propose a new technique for achieving lighting consistency in outdoor real-time AR systems, which does not rely on additional devices and visual information. This method can accurately estimate the sky model and illumination parameters under different lighting conditions in outdoor scenes and meet the real-time requirements of AR systems.

Finally, this paper presents an augmented reality system that achieves good lighting consistency and real-time performance simultaneously, which can be conveniently implemented on mobile devices.

The organization of this paper is as follows. Section 2 reviews related work. Our methods are presented in Section 3. Section 4 describes the experimental setup and discusses the experimental results. A discussion is offered in Section 5.

## 2. Related Work

### 2.1. AR Lighting Estimation Technology

Estimating lighting in augmented reality is an important way to achieve lighting consistency in augmented reality systems. Feng Yan [10] used two photometric spheres as markers to detect brightness changes in the surface of the spheres and thus obtained lighting information from the real environment. Panagopoulos et al. [11] combined 3D information on markers with 2D information on shadows to obtain a series of labels for identifying shadows, and they ultimately estimated the position, shape, and brightness of the light source. Jiddi et al. [22] used specular highlights as auxiliary information. They captured multiple images of scenes containing objects with specular reflections and extracted the positions of specular highlights from the image sequence. After calculating the relationship between the positions of specular highlights in 3D space and the camera's line of sight, they determined the direction of the light source. This method provides accurate light source localization but requires the presence of specular reflection phenomena in the scene. Similarly, Liu et al. [12] used marker and shadow information in images to directly generate virtual object shadows using deep learning without the need for lighting estimation. These marker-based lighting estimation techniques [10–12] require markers with known geometry or material to be present in a scene. Using the shadow and surface image of markers, the lighting information in a real scene can be inferred to achieve lighting consistency in augmented reality. However, this method relies on the presence of markers in an image with known prior information. In addition, when a scene changes, the original information becomes invalid, so this method cannot be applied to real-time dynamic scenes.

In order to obtain lighting information from a scene without the need for markers or prior information, techniques have been developed that use auxiliary devices for lighting estimation [13–16]. Yoo et al. [13] and Pardel et al. [14] used fisheye lenses and stereo cameras, respectively, to capture HDR panoramic images as environment maps, thereby obtaining lighting information from the scene for real-time rendering. Gruber et al. [15] reconstructed a scene model using a depth camera and estimated the lighting of the model using spherical harmonic functions, enabling the generation of soft shadows for virtual objects. Pratul et al. [23] proposed a lighting estimation algorithm that utilizes multi-angle images to generate a panoramic environment map. This algorithm uses paired images from dual cameras as input and uses a CNN to predict the scene beyond the camera view, thus generating an environment map. However, this method requires input in the form of paired images, making it unsuitable for many practical applications. While auxiliary device-based techniques can achieve lighting consistency in augmented reality and attain a certain level of real-time performance, they require specialized equipment for information collection and thus cannot be considered a universal solution for mobile devices.

Techniques that use image analysis for lighting estimation do not require additional specialized equipment and only analyze images captured with the main camera to obtain lighting information from the scene. Karsch et al. [17] estimated a rough geometric model for a scene using images, asked users to annotate the geometry objects and light sources they wanted to interact with, and then rendered virtual objects with lighting consistency accordingly. Lalonde et al. [18] proposed a method that utilizes three clues in the image: sky, vertical surfaces, and ground, to estimate the position and visibility of the sun. Hold-Geoffroy et al. [20] proposed a method that uses deep learning to recover lighting after training a convolutional neural network (CNN) on panoramic images taken outdoors. As a result, a simple HDR map of the sky environment can be directly obtained from low dynamic range (LDR) images captured with the camera as the lighting information of the scene. Zhang et al. [24] proposed an end-to-end network model that incorporates the brightness channel of the sky region as the fourth input channel to estimate the lighting parameters of the scene. Wang et al. [25] utilized a three-dimensional spherical Gaussian representation to model the surface radiance of an entire scene, including both visible surfaces and surfaces outside the field of view. Subsequently, they used standard 3D rendering techniques to render the illumination at any spatial position and viewing angle.

However, image-based techniques have high requirements for image quality and need sufficient clues in the image to perform accurate lighting estimation. Furthermore, these techniques are mostly non-real-time methods, requiring a lot of computation time, and are mainly used for photo composition. Therefore, it is difficult to meet the real-time requirements of AR systems.

As shown in Table 1, most augmented reality lighting estimation techniques can be classified into three categories: marker-based, auxiliary device-based, and image analysis-based. Although these methods have made significant progress in achieving lighting consistency in augmented reality, identifying an accurate and real-time lighting consistency technique remains an important issue in AR visualization. Most of the techniques developed to overcome these issues often require additional devices to collect prior information. These techniques also have difficulties realizing real-time performance in AR systems.

**Table 1.** AR lighting estimation technology.

| Lighting Estimation Technology | Advantage | Limitation |
| --- | --- | --- |
| Based on marker information | Able to accurately estimate lighting information. | Markers or objects need to be set up in advance in the scene. |
| Based on auxiliary equipment | No markers or prior information are needed to be prepared in advance. Lighting information can be estimated in real-time. | Specialized equipment is required to collect information, which makes it difficult to use on mobile devices. |
| Based on image analysis | Lighting estimation can be achieved solely using images without the need for additional specialized equipment. | High image quality is required, and a large amount of computation is needed, which makes it difficult to meet the real-time requirements of AR systems. |

### 2.2. Multi-Source Geographic Information Data

Multi-source geospatial information refers to geographic information obtained from multiple data sources, including but not limited to satellite remote sensing, ground observation, aerial photography, and map production [26]. Geographic information data contains a variety of information that describes spatial features. By fully mining multi-source geospatial information data, we can more accurately describe the lighting information of a scene.

Barreira et al. [27] used GPS data to infer the sun's position to estimate the lighting parameters of a scene. Preetham et al. [28] and Hosek et al. [29] established a sky radiance model based on atmospheric turbidity and surface reflectivity information to describe the lighting information of outdoor scenes. Hold-Geoffroy et al. [20] used convolutional neural networks to construct a sky radiance model, as proposed by Hosek et al. [29], to estimate the lighting of a scene.

Therefore, we can use multi-source geospatial information data to describe a sky radiance model for lighting parameter estimation of outdoor scenes, thereby achieving lighting consistency in AR systems.

## 3. Methods

### 3.1. Overview

To better integrate virtual objects into real-world scenes and achieve a higher level of realism, this research primarily focuses on the influence of lighting and shadows. However, it is important to acknowledge that other factors such as the surface texture of virtual objects and indirect lighting in the scene also play significant roles in the overall realism of the virtual–real fusion.

In this study, our goal is to achieve real-time consistent lighting rendering for augmented reality systems in outdoor scenes using mobile devices without requiring additional equipment or visual information. To achieve this goal, as shown in Figure 1, this study is divided into three parts: integration of multi-source geographic information based on

multi-modal data, generation of a sky radiance model driven using multi-source geographic information, and fusion of real outdoor scenes with virtual 3D objects.

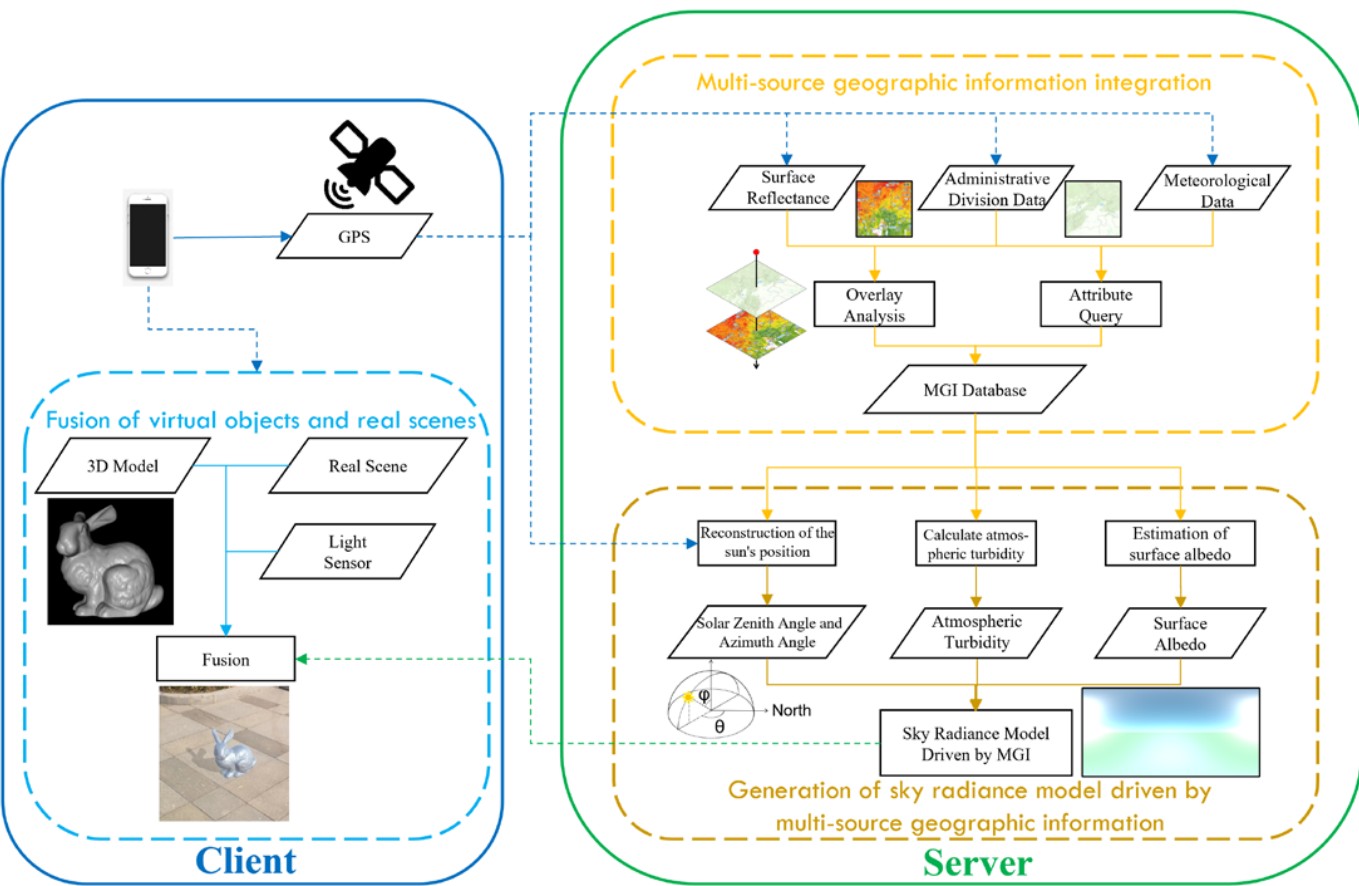

**Figure 1.** Flowchart showing the lighting consistency technique for real-time augmented reality systems in outdoor scenes.

The AR system in this study is divided into two parts: the client side and the server side. As shown in Figure 1, on the client side, we overlay and analyze the latitude and longitude data obtained from mobile sensors with administrative division data and surface reflectance data from the server side, using the results of the overlay analysis to obtain meteorological data for integrating multi-source geographic information. On the server side, we calculate the parameters of the sky radiance model using a multi-source geographic information database and generate the final sky radiance model. Finally, on the client side, we use the sky radiance model and light sensor information to render the outdoor scene's lighting, allowing virtual 3D objects to realistically fuse into real scenes.

### 3.2. Integration of Multi-Source Geographic Information Based on Multi-Modal Data

To accurately describe the lighting information of the target outdoor scene, it is necessary to integrate multi-source geographic information data obtained from the scene. The key operations for integrating multi-source geographic information data are the acquisition, processing, storage, and transmission of multi-modal data from different sensors.

To achieve multi-modal data-based integration of multi-source geographic information, the system is divided into the client side and server side. The client side is responsible for data acquisition and final rendering. On the server side, a multi-source geographic information database is established for data acquisition, processing, and storage. Complex calculations are also performed on the server side to improve the system's FPS (frames per second) and ensure the real-time performance of the client side in the AR system.

Considering the practical needs and application scenarios when using multi-source geographic information data to generate a sky radiance model, the geographic information data integrated into this study includes GPS data, administrative division data, surface reflectance data, and meteorological data.

- GPS data

The GPS (Global Positioning System) is a satellite navigation system developed by the U.S. Department of Defense that provides three-dimensional positioning, velocity, and time information globally. The Global Positioning System consists of three parts: the space segment composed of satellites, the ground control segment, and user receivers. GPS data include information such as satellite position, velocity, and time [30].

- Administrative division data

Administrative division data refer to the division of administrative areas such as provinces, cities, and counties in a country or region, including information such as the names, codes, and boundaries of these administrative regions. In this study, we used GADM (Database of Global Administrative Areas) data as administrative division data. GADM is a high-precision global administrative area database that contains administrative boundary data on multiple levels, such as national borders, provincial borders, city borders, and district borders, for all countries and regions worldwide.

- Surface reflectance data

Surface reflectance, also known as spectral reflectance, refers to the ratio of the reflected flux of an object in a certain wavelength band to the incident flux of that band. It is used to describe the selective reflection of electromagnetic waves at different wavelengths from the surface of objects.

$$\rho(\lambda) = \frac{E_R(\lambda)}{E_I(\lambda)} \tag{1}$$

where $\rho(\lambda)$ is the reflectance of the corresponding wavelength, $E_R(\lambda)$ is the reflected energy of the corresponding wavelength, and $E_I(\lambda)$ is the incident energy of the corresponding wavelength.

In this study, the USGS Landsat 8 Level 2, Collection 2, Tier 12 dataset available on the Google Earth Engine platform was used. This dataset is updated every seven days, undergoes atmospheric correction, and is produced by the Landsat 8 OLI/TIRS sensor. The dataset includes five visible and near-infrared (VNIR) bands, two shortwave infrared (SWIR) bands, and one thermal infrared (TIR) band, which can meet the needs for estimating lighting information.

- Meteorological data

Meteorological data refer to data on various physical quantities used to describe an atmospheric environment, such as temperature, humidity, wind speed, wind direction, air pressure, precipitation, etc. These data are collected using sensors at meteorological stations and can be used in fields such as weather forecasting, climate research, and environmental monitoring.

Considering the need to generate a sky radiance model, this study selected temperature, air pressure, and humidity data as the meteorological data that needs to be integrated.

All the multi-source geographic information data used in this study are shown in Table 2.

After obtaining the required multi-source geographic information data, these data are injected into the server side of the multi-source geographic information database. When users run the AR system on the client side, real-time data acquisition and updates based on a time threshold are conducted according to the GPS information from the client. This provides real-time data support for the system to estimate the lighting information of the scene where the client is located.

**Table 2.** Multi-source geographic information data.

| Data Name | Data Type | Data Source |
|---|---|---|
| GPS data | Text | Mobile device |
| Administrative division data | Vector | GADM (Database of Global Administrative Areas) |
| Surface reflectance data | Raster | Google Earth Engine |
| Meteorological data | Text | China National Meteorological Center |

*3.3. Generation of a Sky Radiance Model Driven Using Multi-Source Geographic Information*

To provide lighting for virtual 3D objects and better integrate them with a real scene, a sky radiance model of the scene is required. To address this issue, this study improves Hosek et al.'s sky radiance model [29] and proposes a new sky radiance model driven using multi-source geographic information.

3.3.1. Reconstruction of the Sun's Position

To establish a sky radiance model driven using multi-source geographic information, an accurate reconstruction of the sun's position is necessary. The attributes that describe the position of the sun include the solar zenith angle and the solar azimuth angle. The solar zenith angle refers to the angle between the incident light direction and the zenith direction, which is the complementary angle of the altitude angle. The solar azimuth angle is generally measured clockwise from the north direction of the target object to the direction of the incident sunlight. As shown in Figure 2, $\theta$ represents the solar zenith angle and $\Phi$ represents the solar azimuth angle.

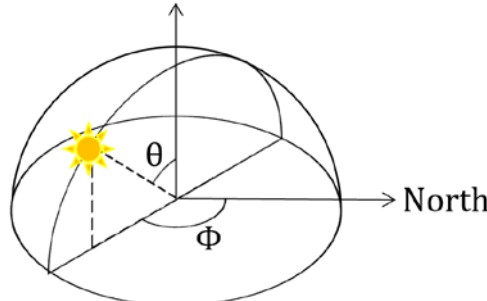

**Figure 2.** Illustration showing the solar zenith angle and solar azimuth angle.

The SPA algorithm [31] is utilized in this study to reconstruct the position of the sun using latitude, longitude, elevation, time, air pressure, and temperature information from the multi-source geographic information database. The current date and time are used to calculate the Julian day ($JD$) with the following equation:

$$JD = INT(365.25 * (T_Y + 4716)) + INT(3.6001 * (T_M + 1)) + T_D + T_B - 1524.5,$$
$$T_B = \begin{cases} 0, & JD < 2299160 \\ 2 - INT(T_Y/100) + INT(T_Y/400), & JD > 2299160 \end{cases} \tag{2}$$

where $T_Y$ is the year, $T_M$ is the month, and $T_D$ is the day of the month with decimal time.

To calculate the $JD$, you should first set $T_B = 0$. If the calculated $JD$ is less than 2,299,160, then the result is correct. However, if the calculated $JD$ is greater than 2,299,160 when $T_B = 0$, then you should update $T_B$ to $(2 - INT\left(\frac{T_Y}{100}\right) + INT\left(\frac{T_Y}{400}\right))$. After updating $T_B$, you can recompute the $JD$ to obtain the correct result.

Once the $JD$ is obtained, the geocentric latitude $\beta$, earth radius vector $R$, true obliquity of the ecliptic $\varepsilon$, apparent sun longitude $\lambda$, and apparent sidereal time at Greenwich $V$ can be determined after combining the Julian day with latitude information and performing a table lookup using the SPA algorithm [31].

The observer local hour angle $H$, geocentric sun declination $\delta$, and equatorial horizontal parallax of the sun $\xi$ are calculated using

$$H = V + \sigma - Arctan\, 2\left(\frac{sin\, \lambda * cos\, \varepsilon - tan\, \beta * sin\, \varepsilon}{cos\, \lambda}\right) \tag{3}$$

$$\delta = Arcsin\,(sin\, \beta * cos\, \varepsilon + cos\, \beta * sin\, \varepsilon * sin\, \lambda) \tag{4}$$

$$\xi = \frac{8.794}{3600 * R} \tag{5}$$

where $\sigma$ is the observer geographical longitude.

After that, the terms $x$ and $y$ can be calculated using the following equations:

$$x = cos\,(Arctan\,(0.99664719 * tan\, \varphi)) + \frac{E}{6378140} * cos\, \varphi \tag{6}$$

$$y = 0.9964719 * sin\,(Arctan\,(0.99664719 * tan\, \varphi)) + \frac{E}{6378140} * sin\, \varphi \tag{7}$$

where $E$ is the observer elevation (in meters) and $\varphi$ is the observer geographical latitude.

After obtaining $x$ and $y$, the parallax in the sun right ascension $\Delta\alpha$ is given by

$$\Delta\alpha = Arctan\, 2\left(\frac{-x * sin\, \xi * sin\, H}{cos\, \delta - x * sin\, \xi * cos\, H}\right) \tag{8}$$

Once the sun right ascension $\Delta\alpha$ is calculated, the topocentric local hour angle $H'$, the topocentric sun declination $\delta'$, and the topocentric elevation angle without atmospheric refraction correction $e_0$ can be obtained using the following equations:

$$H' = H - \Delta\alpha \tag{9}$$

$$\delta' = Arctan\, 2\left(\frac{(sin\, \delta - y * sin\, \xi) * cos\, \Delta\alpha}{cos\, \delta - x * sin\, \xi * cos\, H}\right) \tag{10}$$

$$e_0 = Arcsin\,(sin\, \varphi * sin\, \delta' + cos\, \varphi * cos\, \delta' * cos\, H') \tag{11}$$

Finally, the solar zenith angle $\theta$ and solar azimuth angle $\Phi$ can be calculated using air pressure and temperature data from the multi-source geographic information database and applying the following equation:

$$\theta = 90 - Arcsin\,(sin\, \varphi * sin\, \delta' + cos\, \varphi * cos\, \delta' * cos\, H') - \frac{P}{1010} * \frac{283}{273 + T} * \frac{1.02}{60 * tan\,(e_o + \frac{10.3}{e_o + 5.11})} \tag{12}$$

$$\Phi = Arctan\, 2\left(\frac{sin\, H'}{cos\, H' * sin\, \varphi - tan\, \delta' * cos\, \varphi}\right) + 180 \tag{13}$$

where $P$ is the local pressure and $T$ is the local temperature.

### 3.3.2. Atmospheric Turbidity Calculation

To quantitatively describe atmospheric turbidity, Linke proposed the Linke turbidity factor [32]. The Linke turbidity factor can easily define the appearance of the sky without considering too many meteorological factors. Therefore, it can be used in sky radiance models to describe atmospheric turbidity.

The Linke turbidity factor $TL$ is the ratio of the total optical thickness of the atmosphere $\delta_{atmosphere}$ to the optical thickness of a clean dry atmosphere (CDA) $\delta_{cda}$.

$$TL = \frac{\delta_{atmosphere}}{\delta_{cda}} \tag{14}$$

The total optical thickness of the atmosphere is composed of the effect of CDA, water vapor, and aerosols.

$$\delta_{atmosphere} = \delta_{cda} + \delta_w + \delta_a \tag{15}$$

where $\delta_w$ is the optical thickness of water vapor and $\delta_a$ is the optical thickness of aerosols.

The equation for calculating atmospheric turbidity can be obtained using Equations (14) and (15), as follows:

$$TL = 1 + \frac{\delta_w + \delta_a}{\delta_{cda}} \tag{16}$$

Therefore, once the optical thickness of CDA, water vapor, and aerosols is obtained, the atmospheric turbidity $TL$ can be calculated.

Kasten [33] used optical air mass $m_a$ to estimate the optical thickness of CDA, and the equation is as follows:

$$\delta_{cda} = \left(6.6296 + 1.7513m_a - 0.1202m_a^2 + 0.0065m_a^3 - 0.00013m_a^4\right)^{-1} \tag{17}$$

The absolute air mass $m_a$ is given by Kasten's formula [34]

$$m_a = \frac{P}{1013.25(\cos\,\theta + 0.15(93.885 - \theta)^{-1.253})} \tag{18}$$

where $P$ is the actual pressure in mbar and $\theta$ is the zenith angle.

The optical thickness of water vapor is related to the amount of precipitable water $w$ in the atmosphere. Molineaux [35] proposed an equation to calculate the optical thickness of water vapor based on $w$ and $m_a$.

$$\delta_w = 0.112 - m_a^{-0.55} w^{0.34} \tag{19}$$

The amount of precipitable water can be calculated using Behar's equation [36].

$$w = 0.085 exp\,(2.2572 + 0.05454\,T_{dew}) \tag{20}$$

where $T_{dew}$ is the dew point temperature.

When the temperature $T$ is between 0 °C and 60 °C and the relative humidity $RH$ is between 0% and 100%, the dew point temperature $T_{dew}$ can be calculated using the Magnus–Tetens approximation formula, as follows:

$$T_{dew} = \frac{237.7\eta}{17.27 - \eta} \tag{21}$$

$$\eta = \frac{17.27T}{237.7 + T} + ln\,\left(\frac{RH}{100}\right) \tag{22}$$

The last term, the optical thickness of aerosols, depends on the aerosol transparency coefficient $k$ and optical air mass $m_a$.

$$\delta_a = -\left(\frac{ln\,(k^{m_a})}{m_a}\right) \tag{23}$$

The aerosol transparency coefficient is given by

$$k = 0.9535 - 0.0026w + 7.6927 \times 10^{-4}(90 - \theta) \tag{24}$$

Once the optical thicknesses of CDA, water vapor, and aerosols are calculated, the atmospheric turbidity $TL$ for the current scene can be determined using Equation (16).

### 3.3.3. Surface Albedo Estimation

Surface albedo can significantly affect the brightness of the entire sky [29]. Therefore, in order to obtain an accurate sky radiance model for a given scene, it is necessary to obtain the surface albedo at the location of the scene.

Surface albedo refers to the ratio of reflected radiative flux to incident radiative flux on the surface under solar radiance, reflecting the surface's ability to absorb solar radiance.

$$\alpha = M/E \tag{25}$$

where $\alpha$ represents surface reflectance, $M$ represents reflected radiative flux, and $E$ represents incident radiative flux.

The difference between surface albedo and surface reflectance is that reflectance refers to reflection in a certain direction of a specific band, while albedo is the integral of reflectance over all directions. Reflectance is a function of the wavelength, with different wavelengths having different reflectance. Albedo applies to all wavelengths. Reflectance is used to represent the ratio of reflected energy at a certain wavelength to incident energy. Albedo is used to represent the ratio of reflected energy across the entire spectrum to incident energy.

In this study, the surface albedo was calculated using the inversion model for Landsat data established by Liang [37].

$$\alpha = 0.356B_2 + 0.130B_4 + 0.373B_5 + 0.085B_6 + 0.072B_7 - 0.0018 \tag{26}$$

where $B_2$, $B_4$, $B_5$, $B_6$, and $B_7$ refer to Band SR_B2, Band SR_B4, Band SR_B5, Band SR_B6, and Band SR_B7, respectively, obtained from the USGS Landsat 8 dataset in GEE.

Figure 3 shows surface albedo estimation using the Beijing area as an example. By performing band calculations on the GEE surface reflectance dataset, the surface albedo data are obtained and then used to generate the sky radiance model.

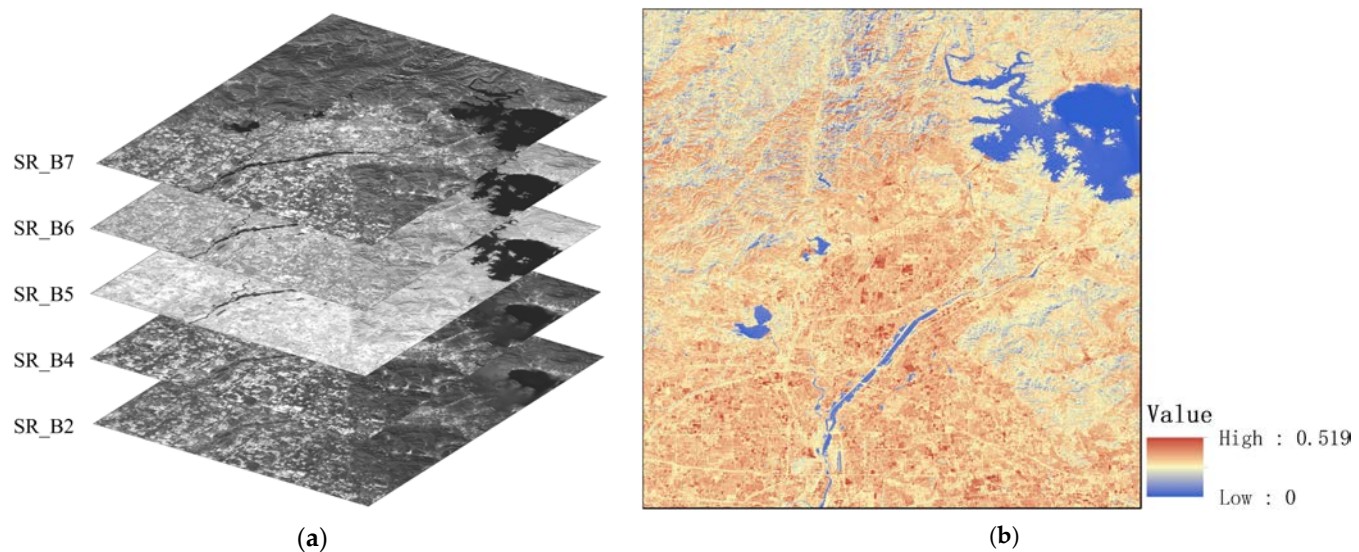

(**a**) (**b**)

**Figure 3.** USGS Landsat 8 surface reflectance dataset and surface albedo in Beijing, 2022. (**a**) USGS Landsat 8 Level 2, Collection 2, Tier 12 Dataset and (**b**) surface albedo.

### 3.3.4. Sky Radiance Model Generation

In order to obtain a sky radiance model for a scene and provide real-world lighting to virtual objects, the sun azimuth, atmospheric turbidity, and surface reflectance data obtained using multi-source geospatial information should be incorporated into the following equation. Then, a sky radiance model driven using multi-source geospatial information can be generated.

$$\mathbb{F}(\theta_V, \gamma) = (1 + Ae^{\frac{B}{\cos\theta_V + 0.01}}) \cdot (C + De^{E\gamma} + F\cos^2\gamma + G \cdot \frac{1 + \cos^2\gamma}{(1 + H^2 - 2H \cdot \cos\gamma)^{\frac{3}{2}}} + I \cdot \cos^{\frac{1}{2}}\theta_V) \tag{27}$$

where $\theta_V$ is the angle between the viewing ray and zenith and $\gamma$ is the angle between the viewing ray and the solar point. The parameters $A$ through $I$ in the equation are used to adjust the brightness distribution, and their calculation equations are as follows:

$$\begin{aligned} v_p^\lambda = \quad & m_{TL,\alpha,p,1}^\lambda \cdot (1-z)^5 + m_{TL,\alpha,p,2}^\lambda \cdot 5z(1-z)^4 + m_{TL,\alpha,p,3}^\lambda \cdot 10z^2(1-z)^3 + \\ & m_{TL,\alpha,p,4}^\lambda \cdot 10z^3(1-z)^2 + m_{TL,\alpha,p,5}^\lambda \cdot 5z^4(1-z)^1 + m_{TL,\alpha,p,6}^\lambda \cdot z^5 \end{aligned} \tag{28}$$

$$z = \sqrt[3]{\frac{90-\theta}{90}}$$

where $p \in \{1, \dots, 9\}$, and the results correspond to parameters $A$ through $I$ in Equation (27), respectively, where $A = v_1^\lambda$, $B = v_2^\lambda$, $C = v_3^\lambda$, $D = v_4^\lambda$, $E = v_5^\lambda$, $F = v_6^\lambda$, $G = v_7^\lambda$, $H = v_8^\lambda$, $I = v_{19}^\lambda$. $\lambda$ is the wavelength of light. After combining the atmospheric turbidity TL and surface reflectance $\alpha$ of the scene, the values $m_{TL,\alpha,p,\{1,\dots,6\}}^\lambda$ in table $M^\lambda$ can be calculated using Hosek's method [29].

Once a sky radiance model driven using multi-source geospatial information is obtained, we can estimate the real-world lighting information of the scene. Figure 4 shows sky radiance models under different levels of atmospheric turbidity and surface reflectance.

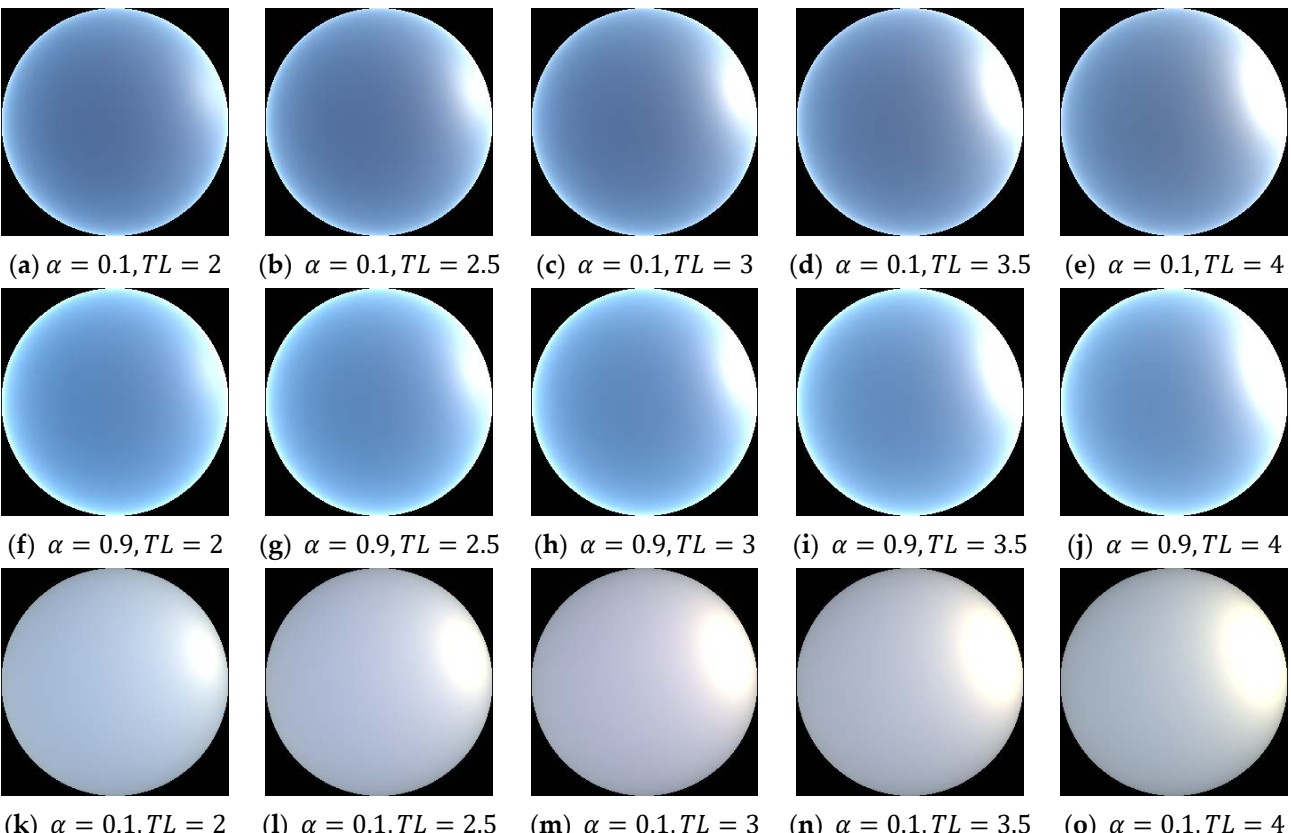

(**a**) $\alpha = 0.1, TL = 2$    (**b**) $\alpha = 0.1, TL = 2.5$    (**c**) $\alpha = 0.1, TL = 3$    (**d**) $\alpha = 0.1, TL = 3.5$    (**e**) $\alpha = 0.1, TL = 4$

(**f**) $\alpha = 0.9, TL = 2$    (**g**) $\alpha = 0.9, TL = 2.5$    (**h**) $\alpha = 0.9, TL = 3$    (**i**) $\alpha = 0.9, TL = 3.5$    (**j**) $\alpha = 0.9, TL = 4$

(**k**) $\alpha = 0.1, TL = 2$    (**l**) $\alpha = 0.1, TL = 2.5$    (**m**) $\alpha = 0.1, TL = 3$    (**n**) $\alpha = 0.1, TL = 3.5$    (**o**) $\alpha = 0.1, TL = 4$

**Figure 4.** Examples of sky radiance models under different conditions. (**a**–**e**) Clear sky radiance model driven using multi-source geographic information data with a surface albedo of 0.1 and atmospheric turbidity ranging from 2 to 4. (**f**–**j**) Clear sky radiance model driven using multi-source geographic information data with a surface albedo of 0.9 and atmospheric turbidity ranging from 2 to 4. (**k**–**o**) Overcast sky radiance model driven using multi-source geographic information data with a surface albedo of 0.1 and atmospheric turbidity ranging from 2 to 4.

### 3.4. Fusion of Real Outdoor Scenes with Virtual 3D Objects

In order to integrate virtual 3D models more realistically with real-world scenes, it is necessary to provide the virtual 3D model with lighting information from the real-world environment. In outdoor scenes, the sun is the most important light source in the environment. Due to atmospheric refraction and scattering, the entire sky emits light, providing indirect daylight illumination for the scene. Therefore, daylighting $E_S$ can be divided into direct sunlight illumination $E_{DS}$ and indirect sunlight illumination $E_{IS}$ [27], as shown in the following equation:

$$E_S = E_{DS} + E_{IS} \tag{29}$$

The sky radiance model driven using multi-source geographic information data in Section 3.3 can obtain indirect sunlight information $E_{IS}$ of the scene using the image-based lighting (IBL) method, which provides indirect illumination for virtual objects. To ensure the real-time performance of the augmented reality system, the model is updated every 20 s or when the change in scene light intensity exceeds 30,000 lux, as the change in indirect sunlight illumination is relatively small.

In order to obtain direct sunlight information $E_{DS}$, it is necessary to determine both the direction and intensity of the direct sunlight represented by the sun. This can be achieved by calculating accurate sun direction information, i.e., the light source position information of direct sunlight, using a combination of Equations (12) and (13), based on GPS information obtained from the client side and the scene multi-source geographic information data on the server side. Then, the intensity of the direct sunlight is obtained by measuring the sunlight intensity data of the scene using the light sensor in a mobile device.

As the light intensity data collected using the light sensor includes both direct and indirect sunlight, it is necessary to establish a relationship between the two types of sunlight. Barreira et al. [27] established a model for the relationship between direct and indirect sunlight intensity under different weather conditions, as shown in Figure 5a–c. However, this illumination model cannot accurately distinguish weather conditions when the scene is under shadows from buildings or clouds. Therefore, we propose an improved model for the relationship between direct and indirect sunlight intensity based on adaptive light intensity, as follows:

$$E_{DS} = \begin{cases} 0.85E_S, & 25000 < E_S \\ 0.5E_S, & 2000 < E_S < 25000 \\ 0, & 0 < E_S < 2000 \end{cases}, E_{IS} = \begin{cases} 0.15E_S, & 25000 < E_S \\ 0.5E_S, & 2000 < E_S < 25000 \\ E_S, & 0 < E_S < 2000 \end{cases} \tag{30}$$

where $E_S$ refers to the overall light intensity of the scene measured in lux using the light sensor. Direct sunlight has a brightness of around 110,000 lux, and the light sensor's upper limit used in our study is set at 90,000 lux. Due to the marginal effect, changes in light intensity have minimal impact on the rendering realism of virtual objects when the light intensity approaches the limit. Therefore, we set the upper limit of the sensor at 90,000 lux as the model limit. The three cases correspond to different light intensity thresholds for weather conditions according to Bird [38], so the model can estimate the illumination under different weather conditions.

As shown in Figure 5d, this method is based on brightness-based scene division and combines the relationship between direct and indirect sunlight intensity under different weather conditions. By adjusting the total light intensity of the scene under different conditions, the model obtains the light source intensity information of direct sunlight.

After obtaining the direct sunlight illumination $E_{DS}$ and indirect sunlight illumination $E_{IS}$ of the scene, the values are inputted into the augmented reality system for real-time rendering of outdoor scenes based on multi-source geographic information data to achieve lighting consistency.

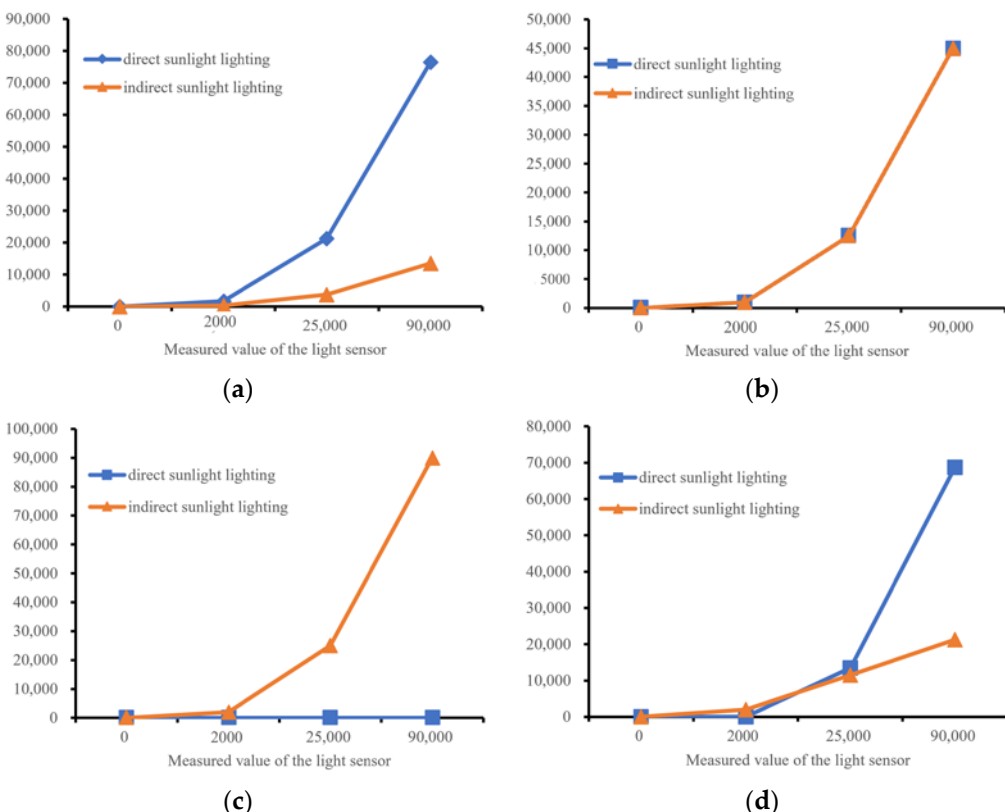

**Figure 5.** Weather-based light intensity model and brightness-based light intensity model. (**a**) Light intensity model for a clear sky. (**b**) Light intensity model for a partly cloudy sky. (**c**) Light intensity model for an overcast sky. (**d**) Light intensity model based on light intensity division.

## 4. Results

To validate the proposed lighting consistency technique for the real-time AR system in outdoor scenes based on multi-source geographic information data, we collected data from multiple outdoor scenes using smartphones at the Aerospace Information Research Institute of the Chinese Academy of Sciences. In the experiment, we compared our technology with several state-of-the-art AR lighting consistency techniques [20,27] and ARCore. We examined the performance of these techniques in terms of the realism of virtual–real fusion and the system frame rate.

### 4.1. Experiments Setup

The prototype device used in this study consists of a server and a client. The server uses Tencent's cloud server, equipped with a 2.6 GHz single-core CPU and 1GB memory, running the Lighthouse lightweight database server. The client uses an Android mobile phone, specifically the Huawei P20 Pro, which is equipped with a Kirin 970 processor, as well as various sensors including a camera, light sensor, and GPS receiver to meet the experimental requirements. This device represents the sensor configuration of most mobile devices on the market, thus ensuring the algorithm's portability across different mobile devices.

To capture HDR lighting information from real-world scenes, we used an Insta 360 ONE X2 panoramic camera in our study. In the same scene, we adjusted the camera's shutter and ISO parameters based on Table 3 to obtain multiple panoramic photos under different exposure conditions. These photos were combined to generate an HDR panoramic photo of the scene, which allowed us to obtain the actual HDR lighting information of the real-world environment. This method takes advantage of the strengths of using a panoramic camera to quickly acquire HDR lighting information from real-world scenes.

**Table 3.** Panoramic camera shooting parameters.

| ISO | Shutter | ISO | Shutter | ISO | Shutter | ISO | Shutter | ISO | Shutter |
|-----|---------|-----|---------|-----|---------|-----|---------|-----|---------|
|     | 1/8000  |     | 1/3200  |     | 1/1250  |     | 1/500   |     | 1/200   |
| 100 | 1/6400  | 100 | 1/2500  | 100 | 1/1000  | 100 | 1/400   | 100 | 1/160   |
|     | 1/5000  |     | 1/2000  |     | 1/800   |     | 1/320   |     | 1/120   |
|     | 1/4000  |     | 1/1600  |     | 1/640   |     | 1/240   |     | 1/100   |

During the development of our AR system, we used the ARFoundation toolkit in Unity, which unifies the ARCore and ARKit development toolkits. The resulting application can be easily deployed to mobile platforms after development completion. This toolkit is capable of satisfying the fundamental requirements of an AR system, facilitating both theoretical research on light consistency techniques and custom development for AR applications.

### 4.2. Evaluation Method and Metric

Similar to the experimental setup in Hold-Geoffroy [20], we utilized the actual HDR lighting conditions from the ground captured in the scene to light the virtual model, using this result as the ground truth (GT) value for the experiment. Then, we compared the realism of the GT value with the predicted results and recorded the system's runtime speed to evaluate the performance of the light consistency techniques in the real-time AR system.

In this study, RMSE (root mean square error), AE (angular error), and SAE (shadow angular error) were used as quantitative evaluation metrics for assessing the effectiveness of virtual–real fusion in AR. These metrics have been widely applied for evaluating light consistency techniques in AR [18,20,39,40]. RMSE is the root mean square error, which measures the overall difference between predicted values and ground truth values by computing the square root of the average of squared differences. It is one of the indicators used to assess image quality. AE refers to the linear RGB angular error, as shown in Figure 6. It measures the accuracy of the predicted results by calculating the angle between the GT and predicted results in the RGB space formed by connecting them to the origin. In this study, when computing RMSE and AE, only shadow positions were considered to prevent alignment errors caused by image capture. SAE measures the angular error between the real and predicted shadows of virtual objects, which can be used to evaluate the accuracy of the light source position and generated shadows.

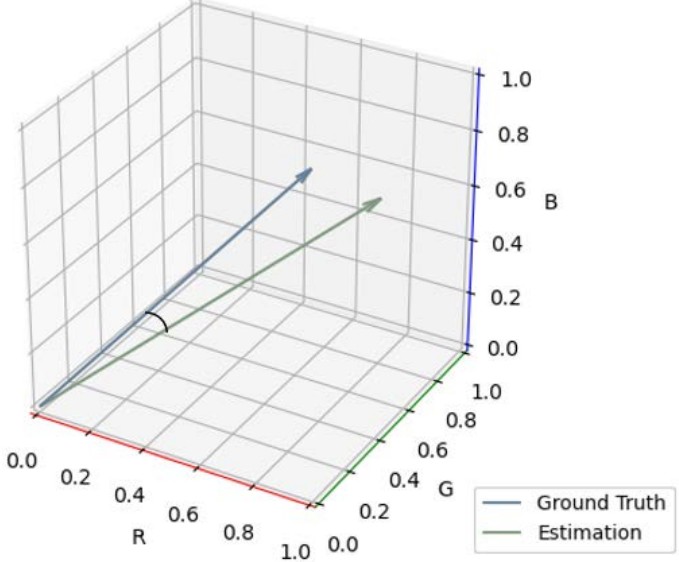

**Figure 6.** The linear RGB angular error.

To assess the real-time performance of the light consistency technique in our AR system, we used Unity's performance analysis tools to monitor the system. By recording the real-time frame rate performance during the operation of the AR system, we evaluated the real-time performance of the light consistency technique in the AR system.

*4.3. Experimental Process and Results*

In order to achieve a realistic fusion of virtual objects with real scenes, it is necessary to use the lighting information of the actual scene to illuminate the virtual objects. In this study, twenty panoramic photos of the scene at different exposures were obtained according to Table 3. These photos were aligned and stitched using the merge-to-HDR tool in Photoshop, resulting in a panoramic HDR with the real lighting texture of the scene. The real lighting of the scene was then used to illuminate the virtual objects with Blender's image-based lighting (IBL) technique, producing accurate shadows and highlights for the virtual objects in the real scene. This result was used as the ground truth (GT) for subsequent experimental comparisons, as shown in Figure 7. To facilitate accurate image capture, the camera was mounted on an adjustable tripod, allowing for repeated and precise captures.

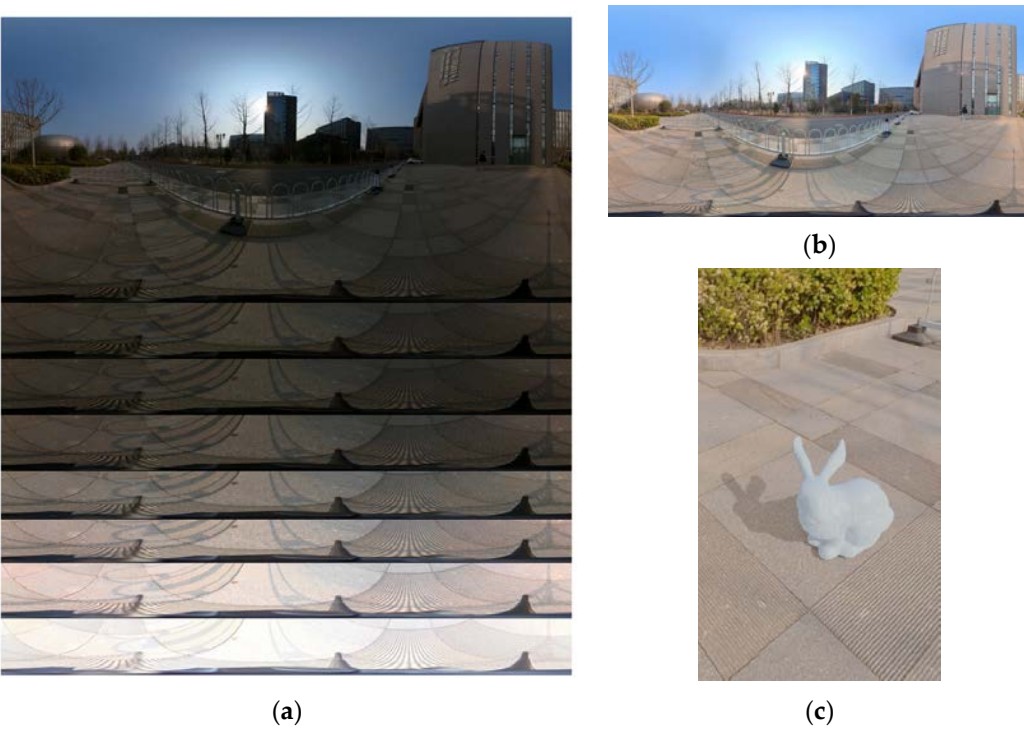

**(a)**　　　　**(b)**　　　　**(c)**

**Figure 7.** The process of obtaining the ground truth. (**a**) Examples of panoramic photos taken at the experimental scene using different exposures. (**b**) Panoramic HDR real lighting texture of the experimental scene. (**c**) Ground truth obtained by illuminating virtual objects using the IBL technique.

To validate our method, we conducted experiments in 20 different scenes and captured panoramic HDR image datasets of the scenes. In each scene, we collected and computed multi-source geographical information data on the scene and used it as input for the model to generate lighting for virtual objects in the AR system within that particular scene.

During our experiments, we compared our method with several state-of-the-art techniques [20,27] and ARCore for achieving AR lighting consistency in outdoor scenes, as shown in Figure 8. We noticed that due to limited lighting and shadow information in the image data, the Hold-Geoffroy and ARCore methods had larger errors in the shadow shape. Our method made significant improvements in the shadow angle and shape when compared to the image analysis methods of Hold-Geoffroy and ARCore, which can be largely attributed to the calculation of the sun's position parameters using multi-source

geospatial information data. Furthermore, our improved lighting brightness estimation resulted in shadow colors that were closer to the ground truth compared to Barreira's method.

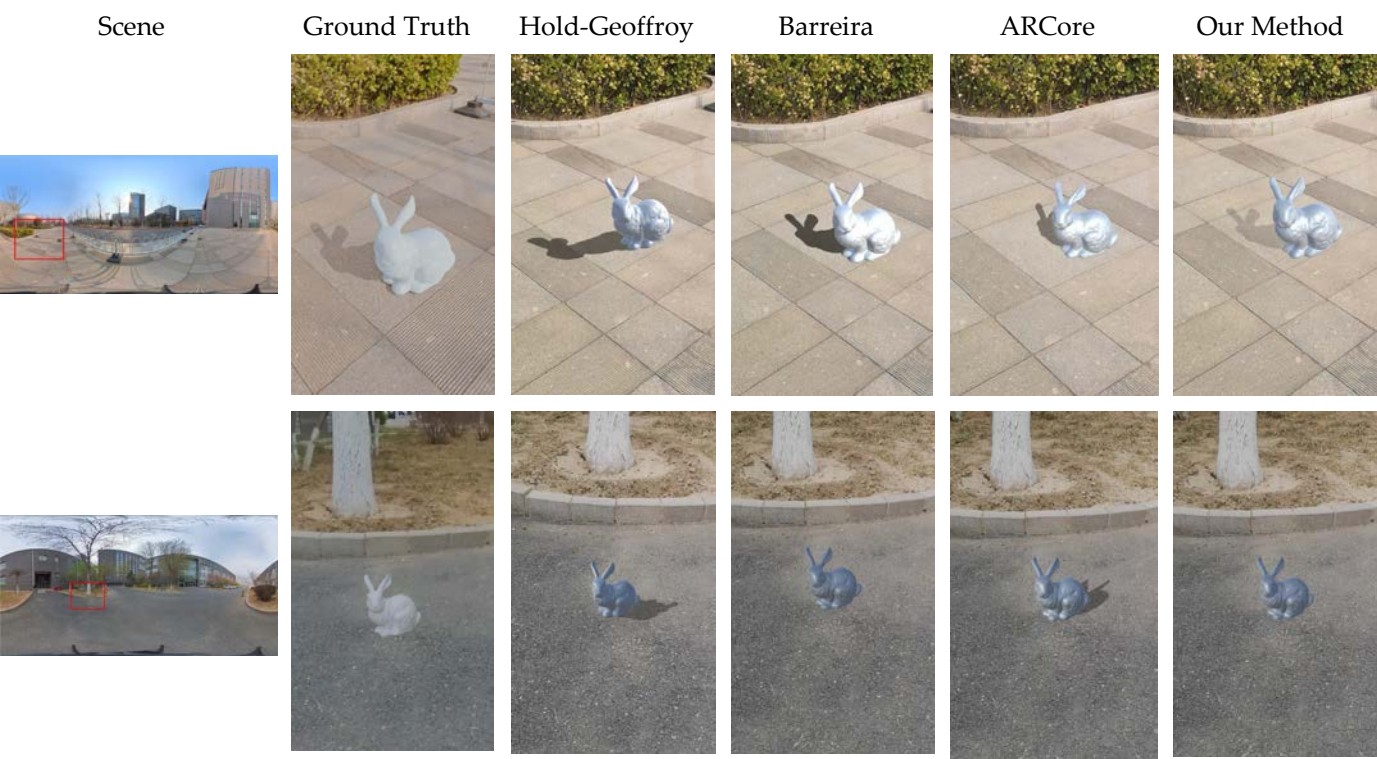

**Figure 8.** The method proposed in this paper are compared with state-of-the-art outdoor AR lighting consistency techniques. The left side displays the HDR panoramic image of the scene, and the rest of the photos show a comparison of the results of different methods with respect to the ground truth.

To quantitatively evaluate the real-time performance of augmented reality systems using different methods, we monitored the system's frame rate during the experiments. Hold-Geoffroy's method can only process images and cannot run in a real-time AR system. The experimental results obtained using other methods are shown in Figure 9. We can see that our method based on multi-source geospatial information does not require complex image analysis work, leading to significantly improved frame rates compared with ARCore. When the client frequently communicates with the server, it reduces the system's frame rate. Therefore, we used a time threshold update method with a resolution of 30 s, which led to a significant improvement in the overall frame rate compared to Barreira's method. Our real-time augmented reality system lighting consistency technology based on multi-source geospatial information can keep the system running at high frame rates, ensuring the real-time performance of the AR system.

To quantitatively evaluate the effectiveness of our method, we used RMSE, AE, and SAE as quantitative evaluation metrics to measure the fusion effect of virtual and real objects in augmented reality. Table 4 shows the overall results of our method and several state-of-the-art outdoor augmented reality lighting consistency techniques in terms of RMSE, AE, SAE, and average frame rates.

From the above experimental results, we can clearly see that the real-time augmented reality system lighting consistency technology based on multi-source geospatial information from outdoor scenes achieves low RMSE and AE, indicating that this method can maintain high realism of a virtual object's shadow colors. At the same time, the achieved SAE of 5.2 indicates that our technology can accurately reproduce the orientation and shape of a virtual object's shadows, significantly improving the realism of the fusion between virtual objects and real scenes. As the method does not require extensive image analysis

calculations, the system's FPS has also been greatly improved, meeting the real-time requirements of AR systems. Moreover, these accuracy and frame rate values align with our claims and demonstrate the value of using multi-source geospatial information in outdoor real-time AR systems.

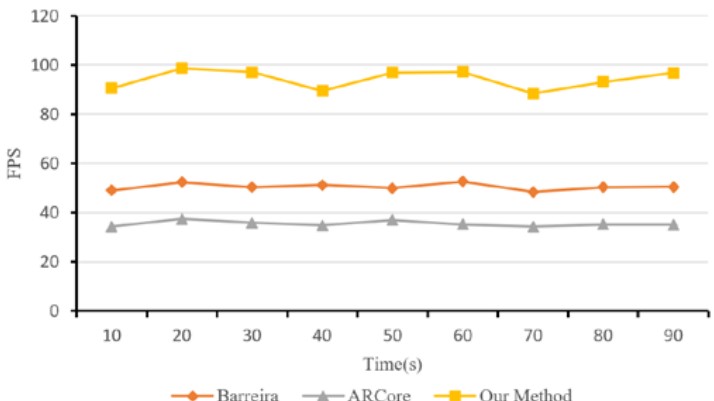

**Figure 9.** System frame rates for different methods.

**Table 4.** A comparison of our method with state-of-the-art techniques for achieving augmented reality lighting consistency in outdoor scenes. The evaluation metrics include the commonly used RMSE, angular error, shadow angular error, and FPS.

|  | Hold-Geoffroy et al. | Barreira et al. | ARCore | Our Method |
|---|---|---|---|---|
| RMSE | 11.1855 | 13.52 | 10.6263 | 10.6547 |
| AE | 0.5208 | 0.5826 | 0.4285 | 0.4927 |
| SAE | 23.8 | 7.3 | 20.3 | 5.2 |
| FPS | — | 50.52 | 35.48 | 94.26 |

## 5. Discussion

The purpose of this paper is to achieve lighting consistency rendering for real-time AR systems in outdoor scenes and optimize the visual quality of virtual object fusion with real scenes. We propose a real-time augmented reality lighting consistency technology based on multi-source geospatial information from outdoor scenes, which overcomes the limitations of scene restrictions and low system frame rates caused by high device requirements, strong image dependence, and a large computational workload in augmented reality lighting consistency techniques. We introduce multi-source geospatial information data and establish a sky lighting model driven using multiple geographic information data elements. Then, we use the sky lighting model and light sensor data to perform lighting rendering for virtual objects in outdoor scenes, enabling them to integrate realistically with the scene. One advantage of our technology is that it can achieve real-time augmented reality lighting consistency rendering without considering additional invasive devices or visual information, ensuring the real-time performance of the system. This makes the technology more easily applicable to a wide range of mobile devices. Experiments conducted at the Aerospace Information Innovation Institute, Chinese Academy of Sciences, demonstrate that our method can achieve high realism for the fusion between virtual objects and real scenes while ensuring the system runs at a high frame rate. In terms of application areas, this method is beneficial for developing AR systems with lighting consistency on mobile devices and has broad prospects in the field of mobile augmented reality.

Compared to methods based on marker information [10–12,22], our approach does not require acquiring prior scene information. Instead, we rely on the analysis of MPI data, enabling us to quickly and accurately obtain outdoor scene lighting information. In contrast to methods based on auxiliary equipment [13–16,23], our approach does not demand extensive specialized sensors. By fully utilizing the sensors available in mobile devices,

we can deploy the algorithm, thus reducing its operational cost. As opposed to methods based on image analysis [17–20,24,25], our approach is independent of images. Instead, we developed a sky radiance model driven using multi-source geographic information, which significantly reduces computational overhead. As a result, we can achieve realistic virtual–real fusion effects at high frame rates.

With the advent of MR devices like Meta Quest Pro and Apple Vision Pro, along with the integration of more advanced and precise sensors in mobile devices, the possibility of conveniently achieving accurate light estimation using auxiliary equipment has become a reality. Light estimation based on auxiliary devices allows for the most precise retrieval of lighting information from a scene. As sensor types, quantities, and accuracy improve, and sensor sizes decrease, coupled with the continuous acceleration of algorithm speeds, light estimation based on auxiliary devices is likely to become the final solution for the research problem o future mobile devices.

However, there are still some limitations. For instance, it is difficult to combine this method with augmented reality lighting consistency techniques for indoor scenes to develop a universal technical solution. Additionally, the accuracy of brightness estimation in our method depends on the quality of the sensors, which can cause errors in the final results using different devices.

**Author Contributions:** Conceptualization, Kunpeng Zhu and Shuo Liu; methodology, Kunpeng Zhu; software, Kunpeng Zhu; validation, Kunpeng Zhu, Yuang Wu and Weichao Sun; formal analysis, Kunpeng Zhu; investigation, Kunpeng Zhu; resources, Kunpeng Zhu; data curation, Kunpeng Zhu and Yixin Yuan; writing—original draft preparation, Kunpeng Zhu; writing—review and editing, Weichao Sun and Shuo Liu; visualization, Kunpeng Zhu; supervision, Weichao Sun; project administration, Shuo Liu; funding acquisition, Shuo Liu. All authors have read and agreed to the published version of this manuscript.

**Funding:** This research was funded by the National Key Research and Development Program of China, grant number 2020YFF0400400.

**Data Availability Statement:** The data presented in this study are available upon request from the corresponding author.

**Conflicts of Interest:** The authors declare no conflict of interest.

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
