# Peer review of "A Lighting Consistency Technique for Outdoor Augmented Reality Systems Based on Multi-Source Geo-Information"

_ijgi, doi:10.3390/ijgi12080324_

Round 1
Reviewer 1 Report
Essentially, this paper is a quality one. The author discusses the calculation of light and shadow effects in the registration of virtual objects in augmented reality from the perspective of light and shadow. This paper reviews the traditional three methods of light and shadow determination, analyzes their advantages and disadvantages, and then puts forward the main research methods of this paper. The method in this paper is essentially the analytical method, that is, the information such as the sun height and incidence angle of a specific scene is determined through the analysis of multivariate geographic information data, and then the rendering strategy of virtual objects is quickly and accurately determined. The paper adopts a combination of qualitative and quantitative methods, the experimental data and process are sufficient, the experimental results are analyzed in place, and the results are credible. However, from the perspective of research logic, the paper has the following problems that need to be clarified: (1) In addition to light and shadow, the registration of virtual objects also needs to be considered by the texture of the physical itself, ambient light and shadow, etc., although it is not within the scope of this article, but obvious problems need to be mentioned; (2) the second type of method, that is, the method of additive sensors, although it is excluded due to efficiency and other issues, but it has to be said that it should be the final solution, because the real world light and shadow effect implies all factors. This is not something that can be covered by general analytical calculations, and although it is currently difficult to achieve due to technical limitations, it cannot be completely ruled out, and it is recommended to elaborate on it in the discussion.
Reviewer 2 Report
More examples may be useful. More references will enrich further the research and state of the art. Further explanations on the equations ( eg. equation 2) are necessary to fully understand the mathematical background
Reviewer 3 Report
The authors introduce a multi-source geographic information technique in order to make lighting more consistent for outdoor AR. They present and compare the results of an evaluation study concerning visualization and computation. This is a very interesting and relevant article.
The article is well written and structured and it was easy for me to follow. I like that the authors clearly stated their contribution at the end of the introduction, which made also the connection to related work easier. The method is well described providing all relevant details. The results suggest that the presented method provides improvements in shadow angle and shape compared to other methods. It is also interesting to see that they achieve such a high number concerning FPS.
I do like this work and see its value. The only critique I have is about the discussion section. The authors should do a better job in describing their results and comparing more thoroughly with the related work. There is a comparison, but rather on high level.
